# Development of a machine learning model for early prediction of plasma leakage in suspected dengue patients

**Ramtin Zargari Marandi** [1]�*, **Preston Leung** [1]�*, **Chathurani Sigera** [2]�, **Daniel Dawson Murray** [1], **Praveen Weeratunga** [2], **Deepika Fernando** [2], **Chaturaka Rodrigo** [3], **Senaka Rajapakse** [2‡], **Cameron Ross MacPherson** [1‡]

**1** Centre of Excellence for Health, Immunity and Infections (CHIP), Rigshospitalet, Copenhagen University Hospital, Copenhagen, Denmark, **2** Faculty of Medicine, University of Colombo, Colombo, Sri Lanka, **3** Viral Immunology Systems Program (VISP), Kirby Institute, UNSW Sydney, Sydney, Australia

577 These authors contributed equally to this work.
‡ These authors are joint senior authors on this work.
* ramtin.zargari.marandi@regionh.dk (RZM); preston.yui.sum.leung@regionh.dk (PL)

## Abstract

### Background

At least a third of dengue patients develop plasma leakage with increased risk of life-threatening complications. Predicting plasma leakage using laboratory parameters obtained in early infection as means of triaging patients for hospital admission is important for resource-limited settings.

### Methods

A Sri Lankan cohort including 4,768 instances of clinical data from N = 877 patients (60.3% patients with confirmed dengue infection) recorded in the first 96 hours of fever was considered. After excluding incomplete instances, the dataset was randomly split into a development and a test set with 374 (70%) and 172 (30%) patients, respectively. From the development set, five most informative features were selected using the minimum description length (MDL) algorithm. Random forest and light gradient boosting machine (LightGBM) were used to develop a classification model using the development set based on nested cross validation. An ensemble of the learners via average stacking was used as the final model to predict plasma leakage.

### Results

Lymphocyte count, haemoglobin, haematocrit, age, and aspartate aminotransferase were the most informative features to predict plasma leakage. The final model achieved the area under the receiver operating characteristics curve, AUC = 0.80 with positive predictive value, PPV = 76.9%, negative predictive value, NPV = 72.5%, specificity = 87.9%, and sensitivity = 54.8% on the test set.

**Data Availability Statement:** All relevant data are within the manuscript and its Supporting Information files. Full data set of this cohort study

is deposited in the institutional repository of the University of Colombo, Sri Lanka and can be accessed for research purposes under the approval of the Human Ethics Research Committee of Faculty of Medicine, University of Colombo (erc@med.cmb.ac.lk). This study relies on open-source tools (S8 Table), particularly R statistical language and mlr3 library with detailed information provided in the Supporting Information. The final model (DENV5f-AS) with an application programming interface (API) is publicly available on https://github.com/PERSIMUNE/ PAC2022Marandi__DengueML_Plasma_Leakage.

**Funding:** This work was supported by the University of Colombo, Sri Lanka (grant no, AP/3/2/ 2017/CG/25), the National Health and Medical Research Council, Australia (Investigator grant no. 1173666 to CR), and the Danish National Research Foundation (DNRF126). The funders had no role in study design, data collection and analysis, decision to publish, or preparation of the manuscript.

**Competing interests:** The authors have declared that no competing interests exist.

## Conclusion

The early predictors of plasma leakage identified in this study are similar to those identified in several prior studies that used non-machine learning based methods. However, our observations strengthen the evidence base for these predictors by showing their relevance even when individual data points, missing data and non-linear associations were considered. Testing the model on different populations using these low-cost observations would identify further strengths and limitations of the presented model.

## Author summary

Current machine learning approaches are mostly designed for decision support systems that used for predicting severity of dengue and forecasting of dengue cases. The few studies to predict plasma leakage rely on traditional statistical approach with a priori predictors. To our knowledge, no study used machine learning to predict plasma leakage in suspected dengue patients. This is the first study to develop a machine learning model to identify predictors that detect plasma leakage in the early stages of dengue infection. In addition, this study placed focus on a limited resource setting enabling decision support systems to be utilised in the low- to middle-income country of Sri Lanka. The study identified five accessible predictors as inputs to the model to aid the decision making for hospital admission for suspected cases of dengue. Lastly, the multi-metric assessments, model fairness and Shapley additive explanations push the agenda for model transparency and interpretability. Our machine learning approach demonstrated reasonable performance and has the potential to identify patients who are likely to develop plasma leakage for close monitoring to prevent severe consequences.

## Introduction

Globally, dengue is one of the fastest growing RNA virus infections with an annual incidence of 390 million infections, and an estimated 10,000 deaths per year [1–4]. Although the disease is endemic in 129 countries, disease burden is mostly in low- and middle-income countries [2,5]. During seasonal outbreaks, healthcare resources of these countries may be overwhelmed by the large number of infections occurring within the span of a few weeks. Therefore, a triaging system to select a subgroup of patients at a higher risk of complications could help prioritise patients needing hospital admission. Almost all people developing life threatening complications of dengue have a "critical phase" of illness characterised by increased capillary permeability and extravasation of plasma (plasma leakage). This leads to underfilling of the circulatory system which if undetected may lead to shock with compromised perfusion to critical organs. While all patients with plasma leakage do not develop life threatening complications, those with complications likely would have had plasma leakage (except for the rare presentation of heavy bleeding in absence of plasma leakage). The estimated portion of subgroup of dengue patients who develop plasma leakage varies between 37–46% of total infections. Identifying this group will capture almost all of the patients at risk of life-threatening complications for monitoring purposes [6]. Plasma leakage is typically observed after day four of fever and therefore a system aiming to predict this outcome should ideally only consider data obtained up to that point. Such a system must also consider data that would be available to a clinician

practicing in resource limited setting, as data requiring complex and expensive laboratory support to generate will have little clinical use in the real-world where it matters.

Sri Lanka is a low-middle income country with a population of approximately 22 million [7]. It had a modest per capita annual health expenditure of USD 157.47 (3.76% of GDP) in 2018 [8]. Seasonal dengue outbreaks result in increased hospitalisations which sometimes overwhelm available resources [9,10]. For instance, in 2017 a disproportionately high number of 186,101 cases of suspected dengue infections were reported with >400 deaths [10]. The direct costs of admitting a single patient suspected to have dengue is between USD 47–68 per hospital stay in Sri Lanka. This is significant given the country's limited healthcare budget [11]. Hence prioritising patients who are most likely to benefit from hospitalisations can be cost saving. For reasons described above, the likelihood of plasma leakage is a good criterion to consider for the admission of patients to hospital.

Machine learning (ML) could be useful to identify patients at risk of complications while allowing early discharge (or non-admission) of "low risk" patients to ensure equitable distribution of limited healthcare resources [10]. Previous studies that attempted to use ML to predict adverse outcomes in dengue showed promise [12,13]. However, these examples have either included laboratory generated data that are not typically available to clinicians in resource limited settings in real-time (e.g., anti-dengue IgG/IgM, viral load, host genetic polymorphisms, single cell RNAseq data [13]), or used an outcome such as "severe dengue" which has limited clinical value because of its subjective definition and low event rate [12,14]. While prediction of the severity of dengue infection has been relatively well documented in either statistical or ML approaches [3,4,12–17], prediction of plasma leakage in comparison, is not [6]. This study addresses the prediction of plasma leakage where the data and the study design are strictly limited to circumstances which on-the-ground clinicians can observe and implement in real-time. For ML model versatility and efficiency, we focus on random forest (RF) [18] and light gradient boosting machine (LightGBM) [19] for their strength and flexibility in handling different types of data and prediction tasks.

## Method

### Ethics statement

Ethical approval for the study was obtained from the Ethics Review Committee, Faculty of Medicine, University of Colombo (EC/17/080) and the Ethics Review Committee, National Hospital of Sri Lanka (ETH/COM/2017/12). All patients had given written informed consent to the donation and storage of serum samples. All methods were performed in accordance with the relevant guidelines and regulations.

### Data collection

The Colombo Dengue Study (CDS) is a prospective cohort study of dengue patients conducted in partnership with Faculty of Medicine, University of Colombo, Sri Lanka and Viral Immunology Systems Program, Kirby Institute, UNSW Sydney, Australia [20,21]. In brief, CDS recruits clinically suspected adult dengue patients from multiple Sri Lankan districts (>12) admitted to the National Hospital of Sri Lanka (in Colombo). Demographic, clinical (signs, symptoms, and laboratory investigation results), socioeconomic data and illness progression (plasma leakage, compensated or uncompensated shock and severe dengue) are recorded with daily follow-up visits until discharge or death. The primary outcome of interest is the occurrence of plasma leakage which is defined as a 20% increment in haematocrit (or a >45% absolute reading) or demonstration of pleural or peritoneal fluid accumulation via ultrasonography. Patient recruitment to CDS began in October 2017 and is ongoing. This analysis focuses on patients recruited

to Phase one of CDS which concluded in February 2020. The diagnosis of dengue in this cohort was confirmed by either a positive NS1 antigen testing or RT-PCR (which also determines viral load and infecting serotype), but the latter was done retrospectively for logistical reasons using stored serum samples obtained within the first four days of fever. However, both these tests are not routinely available for management of patients in Sri Lankan public hospitals and therefore a diagnosis of dengue is often not confirmed in the real-world. Hence mimicking the real-world scenario, all clinically suspected patients with dengue were considered for ML while a subgroup sensitivity analysis (model fairness) was done for the group with laboratory confirmed dengue infections to examine if the predictions would be different compared to the total dataset.

A total of 112 features ("features" and "variables" are used interchangeably to refer to predictors in this study) in each instance was considered (S1 Table). These included sociodemographic data, signs and symptoms of current presentation, laboratory investigation results (haematological, microbiological, and biochemical), imaging results (e.g., ultrasonography findings) and the binary outcome of interest (presence or absence of plasma leakage). Some of these features were "static" as it did not vary across observations (e.g., age) while others were "dynamic" (e.g., blood cell counts). For dynamic features, only the observations up to D3 of fever were considered (D0: first day of observation). For patients with plasma leakage, the time of onset of leakage was estimated as the mid-point between last haematocrit reading which met the 20% increase criterion and the preceding value (serial haematocrit measurements are usually done at 6–8 hourly intervals) or the mid-point between the ultrasound scan which first showed evidence of plasma leakage and the preceding negative scan. If there were no preceding scans or if the time interval between the two scans were >12 hours, the time of plasma leakage was estimated to be 12 hours prior to the positive scan.

The data arrays were summarised using median and interquartile range (IQR) and statistically compared using Wilcoxon rank sum test for continuous variables. For patients with multiple measurements recorded over time (e.g., haemoglobin), the mean was used as a representative value. Categorical variables were statistically compared using Pearson's Chi-square test. In the cohort study with two groups of patients, each patient may have multiple samples to be analysed, leading to multiple hypothesis testing. To control for the risk of false positive results, false discovery rate correction is used to calculate adjusted p-values for multiple testing.

### Data cleaning and selection of features

Both static and dynamic features per patient were merged and linked to a unique patient identification number (Fig 1A). A variable called *observation day* was generated as the number of days since the self-reported onset of fever (D0), to anchor the evolution of dynamic features with a time stamp. Given the multiple observations for dynamic features (e.g., full blood count readings for same patients), the number of instances is higher than the number of patients.

After merging the data sets, instances with missing outcome and post plasma leakage were removed (Fig 1A and 1B). Viral load was removed as a predictor due to the high costs of the procedure in the Sri Lankan hospital setting. Variables from ultrasonography data were similarly removed due to the cost. Alanine aminotransferase (ALT) was also removed due to the high correlation with Aspartate aminotransferase (AST) to avoid redundancy. In addition, constant and semi-constant features with >90% similarity across all instances were removed from the feature set as it would not significantly contribute to the learning process. Finally, instances with more than 50% of the features missing were excluded prior to feature selection to reduce biased interpretation of feature contributions to plasma leakage predictions due to the limitation of extracting feature importance in presence of missingness [22,23].

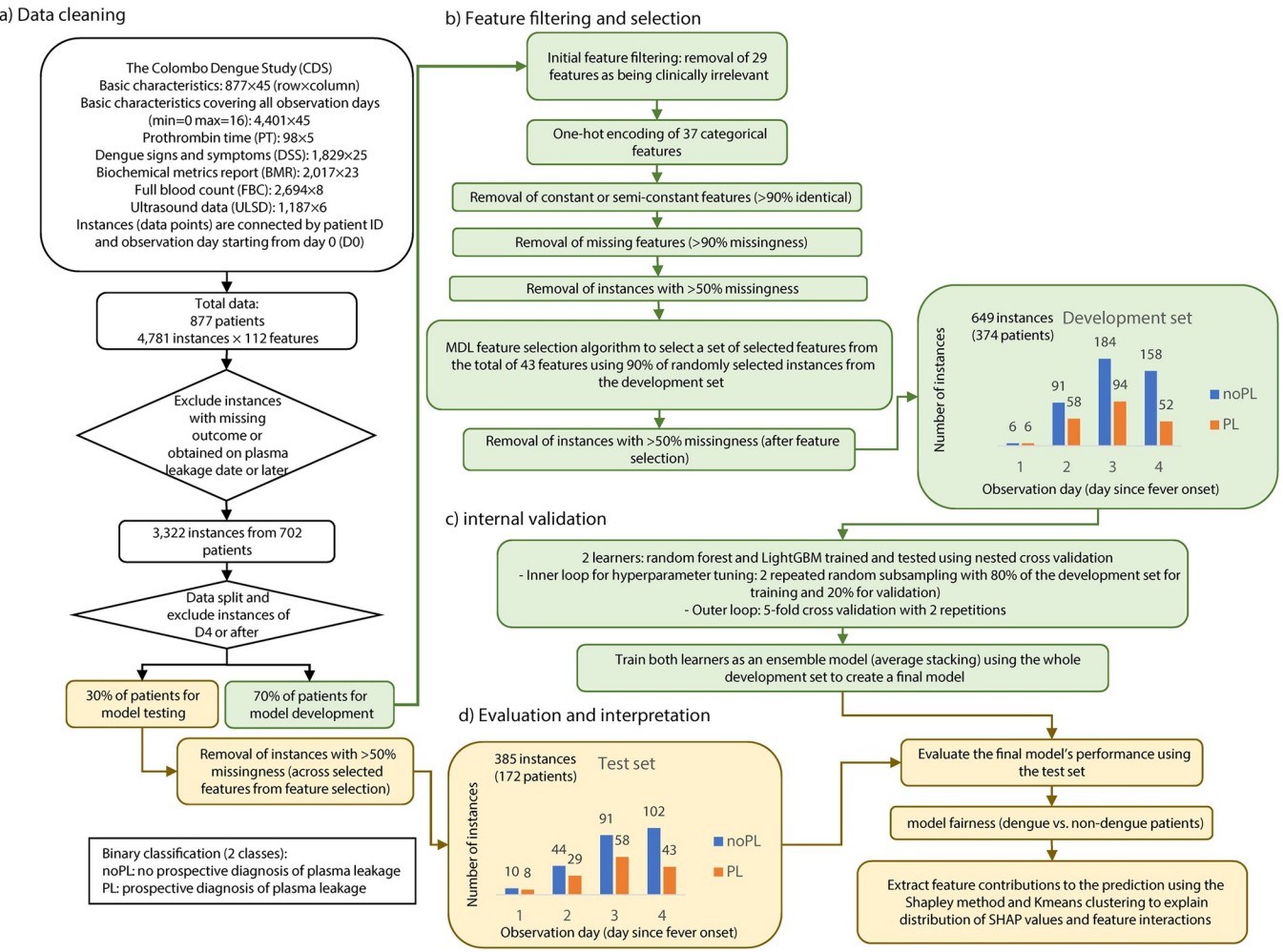

**Fig 1. A summary of the study workflow. a)** Data cleaning, where the datasets in tabular form were merged and instances not meeting the mentioned criteria were removed and the remining instances were split into a development and a test set. **b)** Feature filtering and selection, where the five most relevant features associated with the outcome variable were selected for the learning process. **c)** Internal validation, where two learners compete to predict the occurrence of plasma leakage based on the selected sets of features. The learners were then trained using the whole development set to create an ensemble model based on average stacking as a final model. **d)** Evaluation and interpretation, where the final model is evaluated using the test set and the contribution of the selected features were elucidated using the Shapley method.

The features remaining after the initial filtering (Fig 1B) were fed to the Minimum Description Length (MDL) algorithm [24] to select the most informative features (S2 and S3 Tables). MDL is an information-theoretic method to quantify relative importance of a feature based on its influence on the variability of the outcome. A random subsample consisting of 90% of the development set was selected for feature selection. Different number of selected features were evaluated by prediction performance further in model development and the minimum number of features in which no substantial improvement in prediction performance was observed were selected.

## Prediction of plasma leakage using selected features

The prediction of plasma leakage was defined as a binary classification task with the outcomes (classes) being positive plasma leakage (PL) if the patient was later diagnosed as having plasma

leakage, or not (noPL). The instances were randomly split exclusively by patients to a development set (70% of patients) and a test set (30% of patients).

We trained and evaluated classification models (learners: RF and LightGBM) based on the selected features and the development set (benchmarking experiments) and validate their (prediction) performance on the test set. The learners consist of key hyperparameters (S1 Text) that were tuned using 100 sets in 2 repetitions of bootstrapping over 80% of the instances in the development set to maximise the sensitivity (see S1 Text for description of the metrics).

Nested cross validation was used with 2×5-fold cross validation in outer loop for model evaluation. Thus 10 surrogate models were produced for each learner. The performance for each learner was computed as the average performance of the surrogate models. Averaging of prediction probabilities (i.e., average stacking) was used to integrate LightGBM and RF predictions into a single output. The performance of LightGBM and RF was compared using the Friedman's rank-sum test to confirm the validity of the integration of their outputs in average stacking. The development set was used to train and generate the final model (termed as DENV$_{5F-AS}$. "*5F*" indicates five selected features and "*AS*" for average stacking) which was then evaluated on the test set. The learners generate the probability of each instance belonging to a class (PL or noPL).

The confusion matrix provides proportions of classified instances in each cell of the matrix to the instances in neighbouring cells. The confusion matrix was also generated for a subset of instances to represent a realistic situation where the decision is made using the earliest instances of patient data, thereby providing patient-wise calculations of model performance. Ten different seeds (numbers in computer programming used to initialise a pseudorandom number generation) without replacement (S4 and S5 Tables) were used to assess model reproducibility. The effect of the seed selection on prediction performance was also reported and tested for outliers using Grubbs's test. Furthermore, net benefit of the final model was calculated using decision curve analysis to assess clinical utility of the model [25].

To examine the fairness of the final model, the performance results were assessed via the stratification of confirmed dengue patients and others. The performance metrics for the final model were also reported for each day of fever up to D3 as well as the total performance (average performance across patients). Friedman's rank sum test was used to determine whether significant differences exist between prediction performance across each day of fever. DeLong's test for receiver operating (ROC) curve [26] was applied to identify if a significant difference existed between confirmed dengue patients and others in AUC. Significance level (α) was set to 0.01 for the statistical tests.

Model predictions were subsequently interpreted using the SHapley Additive exPlanations (SHAP) for quantifying feature contributions [27]. The feature contributions in terms of SHAP values were computed for each instance and feature (local importance). The overall contribution of each feature was plotted using SHAP decision plots (global importance) illustrating the interactions between features as a complementary option in our customised implementation. The larger the SHAP value of a feature, the greater the log odds of predicting the sample to be positive for the outcome of interest (i.e., plasma leakage), and vice versa. Interactions between features were illustrated via SHAP decision plots. The global importance in terms of SHAP values are presented as mean absolute SHAP values across correct predictions (matching labels for the outcome of plasma leakage) on the test set. The correlation between SHAP values and feature values were assessed by testing for statistical significance based on Spearman's correlation coefficient (*rho*).

K-means clustering [28] was subsequently performed on SHAP values to identify unique patterns of feature contributing to plasma leakage. K-means clustering is an unsupervised method of assigning instances based on their similarity quantified by a distance matrix to a

chosen number of clusters. The approach allows the identification of unique patterns of feature contributions and their interactions across patients which is otherwise visually challenging when inspected from the original SHAP plot.

Using the K-means clustering, each instance was assigned to a cluster where the SHAP values of the selected features was the closest to its centre as a vector via Euclidean distance (relative to neighbouring clusters). Four clusters were selected to distinguish the patterns of feature interactions in the SHAP plot. The clustering parametrisation results are also provided.

To evaluate model consistency, abstention windows were applied to the predictions on the test set to assess the prediction performance by enforcing restrictions on acceptable ranges of predicted probabilities to be assigned to classes [29], see S1 Text.

## Result

A total of 877 patient records were available from Phase one of CDS (69.4% male; age in years: median = 30, IQR = 22–45) each consisting of datapoints for up to 112 static and dynamic features. A total of 529 (60.3%) patients had confirmed dengue infection. Plasma leakage was observed in 346 patients (39.5%), not observed in 359 (40.1%) patients, and not recorded in 172 (19.6%) patients. Descriptive data of the cohort is summarized in Table 1. The variable with the highest missingness was Serum total bilirubin (54.2% in total) and the variable with the greatest difference in missingness (11.4%) between dengue and non-dengue groups was BMI.

The data was split to a development set comprising of 374 patients and a test set with 172 patients (Fig 1). In the development set, 43 of the initial set of 112 features (S2 Table) satisfied the filtering criteria as illustrated in Fig 1B. MDL was applied to these 43 features to select the top five features that were most informative for the prediction of plasma leakage (S3 Table). The selected features were HCT, HGB, AST, lymphocyte count, and age recorded up to D3 of fever. In the development set the outcome of plasma leakage was observed in 131 (35%) patients while in the test set it was observed in 73 (42%) patients.

### Model development and validation

The benchmarking results of the two learners as well as their average stacking (Ensemble) using the development set are summarised in Table 2. The learners performed well according to their single-snapshot performance metrics (MCC, BA, PPV, NPV, sensitivity, and specificity) compared to the theoretical chance level and the mean Brier score of $\leq 0.15$.

The performance of the learners remained consistently higher than the chance level (diagonal dashed line in Fig 2A and horizontal dashed line in Fig 2B) for both AUC and PRAUC in different probability thresholds exhibiting robustness in predictions (Fig 2A and 2B). The performance of the final model ($DENV_{5F\text{-}AS}$) on the test set is presented in ROC and precision-recall (PR) curves in Fig 2C and 2D, respectively. The model performed on the test set had similar performance to the learners during the benchmarking experiments.

The performance of the final model $DENV_{5F\text{-}AS}$ on the test set is presented in Fig 2C–2F. The ROC and PR curves are above the chance level (dashed lines in Fig 2C–2D) with AUC = 0.80 and PRAUC = 0.69. The model exhibited specificity = 89.1%, sensitivity = 50.0%, PPV = 71.9% and NPV = 76.1% as indicated in the confusion matrix in green (Fig 2E). The prediction performance when considering only the earliest instance of the patient data (confusion matrix in orange–Fig 2E) had specificity = 87.9%, sensitivity = 54.8%, PPV = 76.9% and NPV = 72.5%. The regression line to the pool of predicted probabilities over the observation days revealed no statistical significance of descending or ascending trends ($R^2 < 0.1$, Fig 2F). The results regarding the model consistency by random selection of samples (S4 and S5

**Table 1. Summary statistics of cohort grouped by dengue and non-dengue patients.**

| Feature/variable | Non-dengue N = 348[a] | Dengue N = 529[a] | Total N = 877[a] | q[b] |
|---|---|---|---|---|
| Duration of fever (days) | 5.00 (1.75, 6.00) | 6.00 (5.00, 7.00) | 5.00 (4.00, 7.00) | <0.001 |
| Age (years) | 36 (24, 52) | 27 (21, 39) | 30 (22, 45) | <0.001 |
| (Missing data) | 0 (0.0%) | 1 (0.2%) | 1 (0.1%) | |
| Gender (male) | 250 / 348 (72%) | 359 / 529 (68%) | 609 / 877 (69%) | 0.3 |
| BMI (kg/m$^2$) | 22.2 (19.6, 25.1) | 21.7 (19.1, 24.7) | 21.9 (19.2, 24.8) | 0.2 |
| (Missing data) | 158 (45.4%) | 180 (34.0%) | 338 (38.5%) | |
| Plasma Leakage (positives) | 86 / 259 (33%) | 260 / 446 (58%) | 346 / 705 (49%) | <0.001 |
| (Missing data) | 89 (25.6%) | 83 (15.7%) | 172 (19.6%) | |
| Dengue virus serotype | | | | |
| DENV1 | - | 50 / 426 (12%) | 50 / 426 (12%) | - |
| DENV2 | - | 257 / 426 (60%) | 257 / 426 (60%) | - |
| DENV3 | - | 107 / 426 (25%) | 107 / 426 (25%) | - |
| DENV4 | - | 12 / 426 (2.8%) | 12 / 426 (2.8%) | - |
| (Missing data) | - | 103 (19.5%) | 103 | |
| Viral Load (PFU/mL) | - | 22688 (3536, 270520) | 22688 (3536, 270520) | - |
| (Missing data) | - | 103 (19.5%) | 103 | |
| Haemoglobin [c] (g/dL) | 12.79 (11.51, 14.30) | 13.58 (12.02, 14.40) | 13.37 (11.80, 14.34) | 0.018 |
| (Missing data) | 88 (25.3%) | 83 (15.7%) | 171 (19.5%) | |
| Haematocrit [c] (%) | 40.7 (36.6, 44.3) | 42.0 (37.9, 44.5) | 41.5 (37.4, 44.5) | 0.008 |
| (Missing data) | 88 (25.3%) | 83 (15.7%) | 171 (19.5%) | |
| Leukocyte count [c] (10$^3$/μL) | 5.46 (3.85, 7.62) | 3.47 (2.76, 4.57) | 3.98 (3.02, 5.79) | <0.001 |
| (Missing data) | 88 (25.3%) | 83 (15.7%) | 171 (19.5%) | |
| Platelets count [c] (10$^3$/μL) | 124 (91, 164) | 83 (55, 116) | 98 (64, 134) | <0.001 |
| (Missing data) | 88 (25.3%) | 83 (15.7%) | 171 (19.5%) | |
| Neutrophil count [c] (10$^3$/μL) | 2.62 (1.62, 4.51) | 1.64 (1.19, 2.25) | 1.82 (1.29, 2.85) | <0.001 |
| (Missing data) | 88 (25.3%) | 83 (15.7%) | 171 (19.5%) | |
| Lymphocyte count [c] (10$^3$/μL) | 1.46 (1.04, 2.03) | 1.18 (0.89, 1.70) | 1.28 (0.94, 1.79) | <0.001 |
| (Missing data) | 88 (25.3%) | 83 (15.7%) | 171 (19.5%) | |
| AST [c] (U/L) | 38 (29, 63) | 67 (39, 111) | 54 (34, 97) | <0.001 |
| (Missing data) | 114 (32.8%) | 102 (19.3%) | 216 (24.6%) | |
| ALT [c] (U/L) | 38 (22, 62) | 51 (29, 90) | 46 (27, 83) | <0.001 |
| (Missing data) | 114 (32.8%) | 103 (19.5%) | 217 (24.7%) | |
| Serum sodium [c] (mmol/L) | 136.0 (134.0, 138.0) | 136.0 (134.0, 137.9) | 136.0 (134.0, 138.0) | 0.3 |
| (Missing data) | 113 (32.5%) | 106 (20.0%) | 219 (25.0%) | |
| Serum potassium [c] (mmol/L) | 3.80 (3.60, 4.19) | 3.80 (3.50, 4.10) | 3.80 (3.50, 4.15) | 0.3 |
| (Missing data) | 113 (32.5%) | 106 (20.0%) | 219 (25.0%) | |
| Serum creatinine [c] (μmol/L) | 80 (66, 94) | 79 (66, 92) | 79 (66, 93) | 0.3 |
| (Missing data) | 110 (32.0%) | 105 (19.8%) | 215 (24.5%) | |
| C-reactive Protein [c] (mg/L) | 18 (7, 52) | 12 (6, 26) | 15 (6, 31) | <0.001 |
| (Missing data) | 115 (33.0%) | 123 (23.2%) | 238 (27.1%) | |
| Serum total bilirubin [c] (μmol/L) | 7 (1, 12) | 6 (1, 11) | 6 (1, 12) | 0.3 |
| (Missing data) | 208 (60.0%) | 267 (50.5%) | 475 (54.2%) | |

(a)—Median (IQR) for continuous variables; n / N (%) for categorical variables.

(b)—False discovery rate correction for multiple testing for Wilcoxon rank sum test; Pearson's Chi-squared test.

(c)—Each patient had multiple measurements of this data over time. The mean of the measurements was taken within-patient as a representative value prior to calculating the median, IQR and testing over all patients.

**Table 2. Benchmarking results for random forest (RF), gradient boosting machine (LightGBM), and their average stacking (Ensemble) using the nested cross validation on the development set.**

| Learner | MCC | BA | PPV | NPV | Specificity | Sensitivity | AUC | PRAUC | Brier |
|---|---|---|---|---|---|---|---|---|---|
| RF | 0.51±0.09 | 0.72±0.04 | 0.78±0.09 | 0.80±0.04 | 0.93±0.04 | 0.52±0.06 | 0.85±0.05 | 0.76±0.07 | 0.14±0.02 |
| LightGBM | 0.49±0.07 | 0.73±0.03 | 0.69±0.08 | 0.82±0.04 | 0.87±0.03 | 0.59±0.06 | 0.84±0.03 | 0.75±0.06 | 0.15±0.02 |
| Ensemble | 0.53±0.05 | 0.73±0.02 | 0.79±0.09 | 0.81±0.04 | 0.93±0.03 | 0.53±0.05 | 0.86±0.03 | 0.78±0.05 | 0.14±0.01 |

AUC: area under the receiver operating characteristics curve, PRAUC: area under the precision-recall curve, balanced accuracy BA = ((TP/(TP+FN)+TN/(TN+FP)))/2, negative predictive value (NPV) = TN/(TN+FN), positive predictive value (PPV) = TP/(TP+FP), sensitivity = TP/(TP+FN), specificity = TN/(TN+FP), Matthews correlation coefficient (MCC) = (TP×TN-FP×FN)/$\sqrt{}$(TP+FP)(TP+FN)(TN+FP)(TN+FN), Brier = mean squared error between predicted probabilities and the observed values (plasma leakage = 1, no plasma leakage = 0). TP, TN, FP, and FN are true positives, true negatives, false positives, and false negatives, respectively. The positive class is being clinically diagnosed as plasma leakage using polymerase chain reaction test. The values for the performance measures indicate mean ± standard deviation of surrogate models from 2×5-fold cross validation in the outer loop.

Tables) and applying abstention windows are described in S1 Text. In addition, S1 Fig depicts net benefit of the proposed model (DENV$_{5F-AS}$) throughout the range of probability thresholds, indicating that the model is of clinical value as compared to baseline (either random guess or classifying all samples as PL).

The prediction performance of the final model (DENV$_{5F-AS}$) on the test set is summarised in Table 3. DENV$_{5F-AS}$ achieved the total performance of AUC = 0.80. The model performance varied within the range of [0.69 0.84] in AUC representing the robustness of the model across D0 to D3 post onset of fever. The Brier score as a model calibration measure was ≤0.22. MCC was 24% higher in the non-dengue group (MCC = 0.47) compared with the dengue group (MCC = 0.38).

## Model fairness

ROC plots for DENV$_{5F-AS}$ illustrated the stratified groups (dengue vs non-dengue) on the test set (Fig 3A). The model performance did not differ significantly for the dengue group (AUC = 0.74) compared to the non-dengue group (AUC = 0.87). The confusion matrix of DENV$_{5F-AS}$ is depicted in Fig 3B for the dengue and non-dengue groups from the test set. It shows that the model performed better in predicting PL when tested on the dengue group (PPV = 75.7%) whereas it performs better in predicting noPL when tested on the non-dengue group (NPV = 90.2%). Similar results were achieved when only considering the earliest instance of each patient PPV = 78.0% for dengue and NPV = 87.5% for non-dengue group (Fig 3C).

## Interpretability of the model

The SHAP decision plot in Fig 4 shows the contribution and interaction of selected features to the predicted probabilities for plasma leakage. Each group of points directed by coloured lines across features in Fig 4 represents an instance. Lymphocyte count had the highest contribution to the prediction of plasma leakage with mean absolute SHAP value of 0.091. It was negatively correlated with its SHAP values (Spearman's *rho* = -0.91, p<0.001) that translates to decreased predicted risk of plasma leakage as it increased. HGB was the second most contributing feature with its values being positively correlated with its SHAP values (Spearman's *rho* = 0.84, p<0.001) reflecting increased predicted risk of plasma leakage as it increased. HCT as the third most contributing feature followed a similar pattern with less strength in correlation (Spearman's *rho* = 0.34, p<0.001) where its higher values contributed to increased predicted risk of plasma leakage and as it decreases to its lower values the contribution approached the

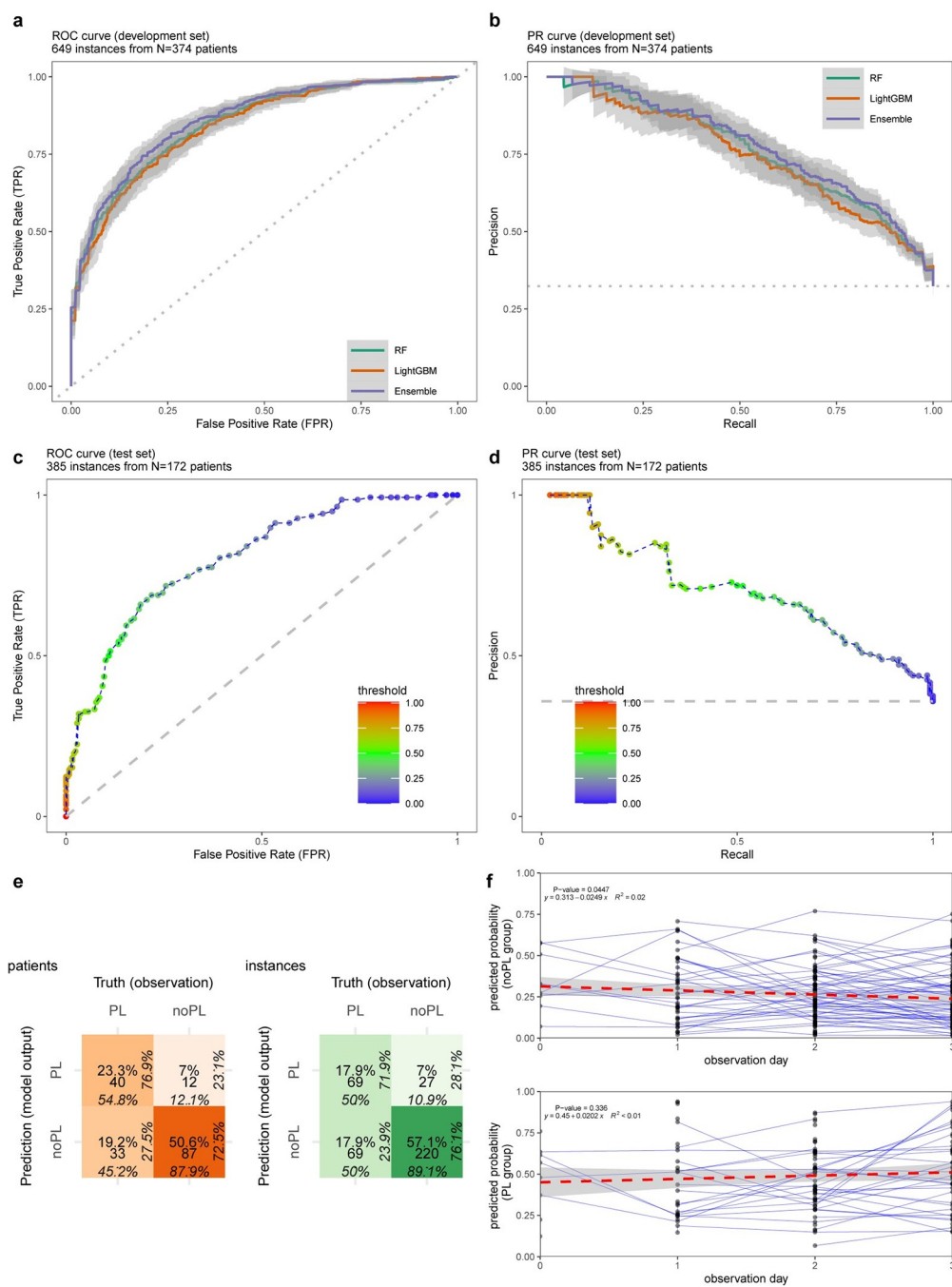

**Fig 2. An overview of the machine learning results. a)** ROC curve for gradient boosting machine (LightGBM), random forest (RF) and their average stacking (Ensemble) using varying prediction probability thresholds from 0 to 1 (step size = 0.01). **b)** PR curves for the learners using the same thresholding. **c)** ROC for the final model, $DENV_{5F\text{-}AS}$: 5-Featured Average Stacking of LightGBM and RF, on the test set (AUC = 0.80). **d)** PR curve for the final model on the test set (PRAUC = 0.69). **e)** Confusion matrix for $DENV_{5F\text{-}AS}$ on the test set (green: all instances, orange: earliest instances of patients), percentages in each direction provide the proportion of instances of each row or column. PL: plasma leakage and noPL: no plasma leakage. The colour intensity is proportional to the ratio of the instances of a matrix cell to the total number of instances. The percentages indicate proportion of classified instances in each cell to the instances in neighbouring cells by row or column, **f)** Prediction probabilities during the observation period for patients in each class where each point indicate an instance and the connected points indicate that the instances belong to the same patient. The trends of the predicted probabilities for each class are shown by fitting linear regression.

**Table 3. Prediction performance of the final model to predict plasma leakage on test set.**

| Model (DENV$_{5F-AS}$) | MCC | BA | PPV | NPV | Sensitivity | Specificity | AUC | PRAUC | Brier |
|---|---|---|---|---|---|---|---|---|---|
| Performance on the test set (172 patients, 385 instances) | 0.43 | 0.70 | 0.72 | 0.76 | 0.50 | 0.89 | 0.80 | 0.69 | 0.17 |
| *By dengue and non-dengue* | | | | | | | | | |
| dengue (105 patients, 241 instances) | 0.38 | 0.68 | 0.76 | 0.66 | 0.50 | 0.86 | 0.74 | 0.72 | 0.21 |
| non-dengue (67 patients, 144 instances) | 0.47 | 0.72 | 0.59 | 0.90 | 0.52 | 0.92 | 0.87 | 0.61 | 0.11 |
| *By day of post fever onset* | | | | | | | | | |
| Day 0 | 0.20 | 0.60 | 0.57 | 0.64 | 0.50 | 0.70 | 0.69 | 0.60 | 0.22 |
| Day 1 | 0.38 | 0.67 | 0.70 | 0.72 | 0.48 | 0.86 | 0.72 | 0.62 | 0.20 |
| Day 2 | 0.48 | 0.71 | 0.81 | 0.74 | 0.50 | 0.92 | 0.82 | 0.74 | 0.17 |
| Day 3 | 0.44 | 0.70 | 0.67 | 0.81 | 0.51 | 0.89 | 0.84 | 0.69 | 0.14 |

midline reflecting less impacts on the predictions. The fourth most contributing feature was age (mean absolute SHAP = 0.050) where the younger mostly contributed to increased predicted risk of plasma leakage and vice versa (Spearman's *rho* = -0.57, p<0.001). Lastly, AST increased the predicted risk of plasma leakage when its values were mostly higher than the average in the range and vice versa (Spearman's *rho* = 0.85, p<0.001).

Visually distinct groups of interactions across features illustrated as edges connecting points denote patterns. The lines connecting features for each instance in Fig 4 additionally showcased the complexity to follow the different patterns for contributing to the outcome of plasma leakage. To explore the patterns showcased in Fig 4 and whether distinct subgroups of patients were contributing to the prediction of plasma leakage, clusters were derived by running the K-means clustering algorithm based on their distinguishing feature values. Briefly, four clusters were identified: Cluster 1 included 148 (62) instances (patients), Cluster 2–106 (44), Cluster

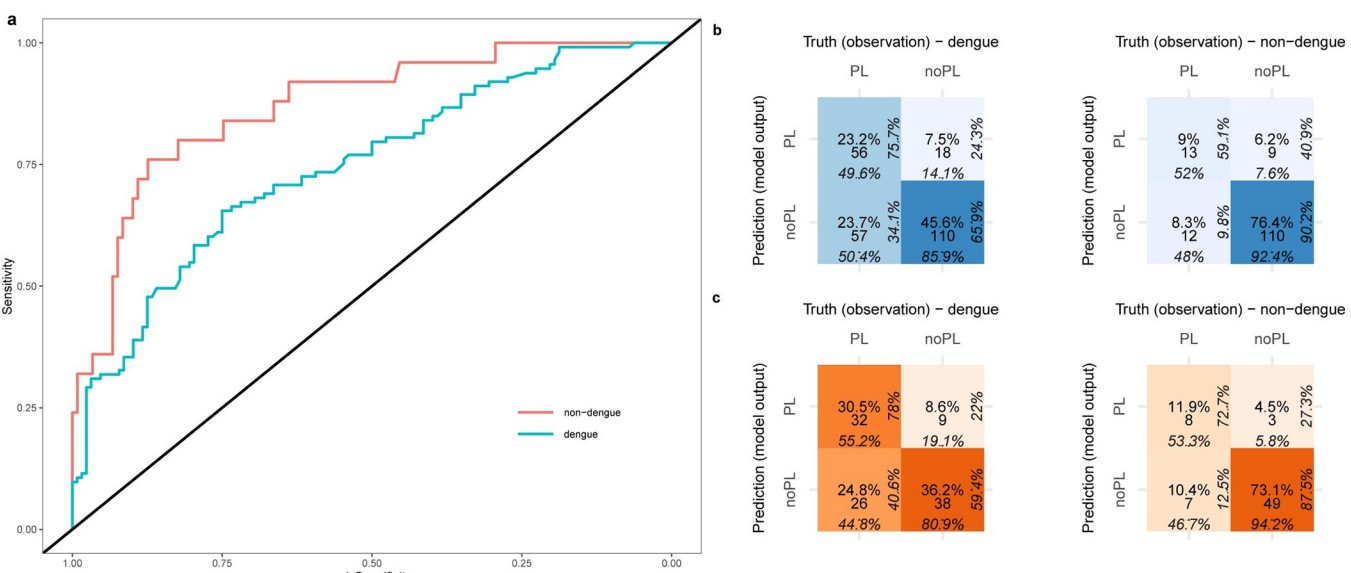

**Fig 3. An overview of the model fairness of the final model: DENV$_{5F-AS}$. a)** ROC curves for the dengue and Non-dengue subsets of the test set. **b)** Confusion matrix (in blue) for the DENV$_{5F-AS}$ on the dengue and non-dengue subsets of the test set, percentages in each direction provide the proportion of instances that belong to each row or column of the confusion matrix. PL: plasma leakage and noPL: no plasma leakage. The colour intensity is proportional to the ratio of the instances of a matrix cell to the total number of instances, **c)** confusion matrix (in orange) only including the earliest instance of patients for dengue and non-dengue subsets of the test set.

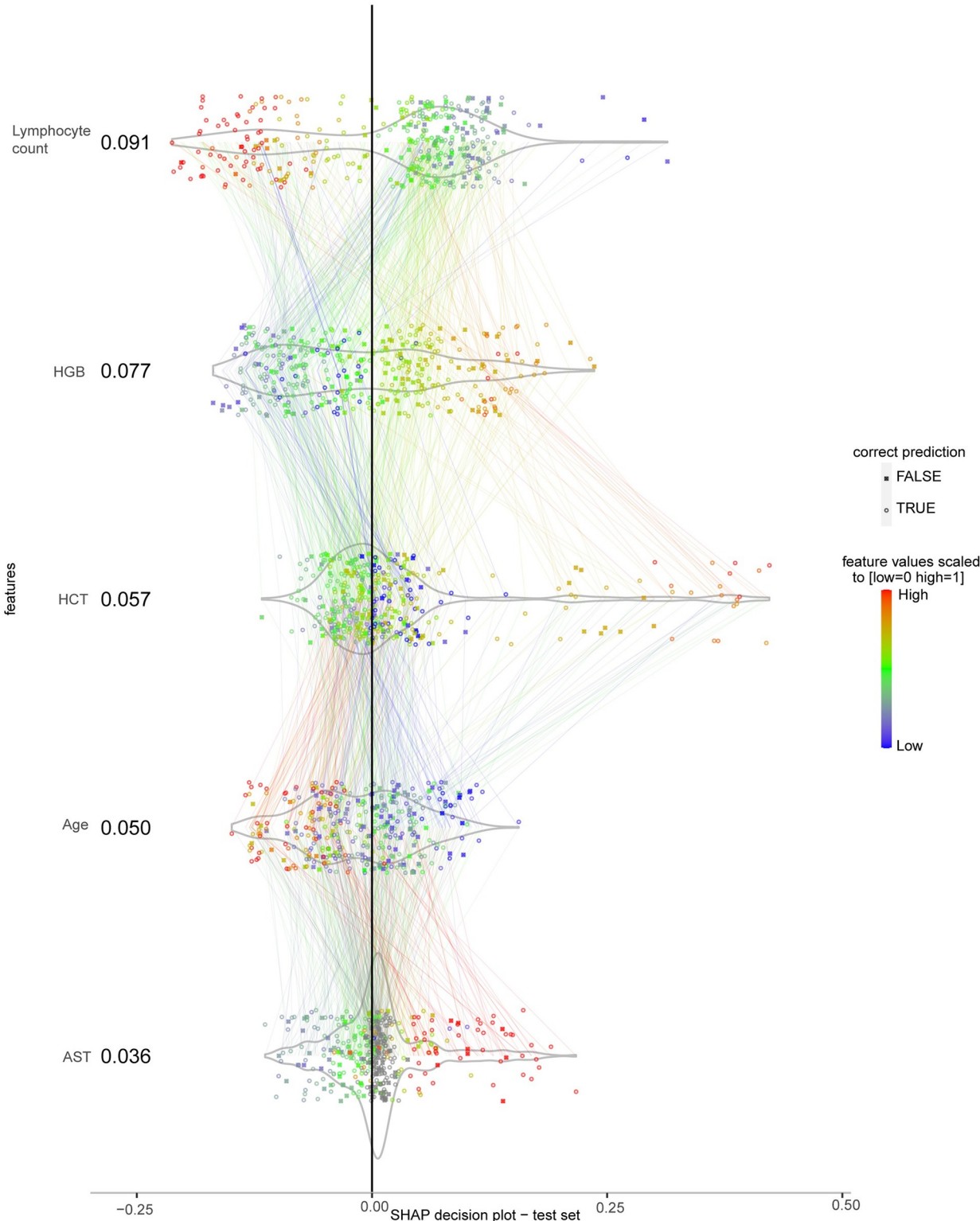

**Fig 4. SHAP decision plot for DENV$_{5F-AS}$ on the test set for correct and incorrect predictions.** The features are sorted from top to bottom by their mean absolute SHAP values in correct predictions. Each point represents an instance and the connected lines across features belong to the same instance. For each feature the points are scattered perpendicular to the horizontal line to minimise overlapping. Feature values are normalised to [0 1] by the min-max normalisation method and colour-coded (grey points are missing values), outliers were squished to the range using Hampel filter. The colour of each line is the same as the value of the feature connected to in downwards direction. X-axis is the SHAP value

computed for each instance in terms of log-odds of the predicted plasma leakage probabilities. The plot is centred on the x-axis at the baseline level of SHAP value determined by the algorithm.

3–37 (22), and Cluster 4–94 (34). The four clusters illustrated that each subgroup contributed differently to the prediction performance (see S2 Fig and S6 Table)

## Discussion

We present an interpretable ML approach to predict the occurrence of plasma leakage in patients clinically suspected of dengue fever using real-world data observed within the first 96 hours of fever from a resource limited setting. The final model implemented using the ensemble of RF and LightGBM, was found effective in early prediction of plasma leakage for the dengue and non-dengue patients. The model had high specificity in predicting plasma leakage but lower sensitivity, indicating that its useful to identify patients needing close monitoring upfront, but less capable of determining those who can be safely discharged.

This study only used data that are typically available to clinicians managing patients even in peripheral hospitals in Sri Lanka. It is a realistic situation where similarities could apply likely in other resource limited countries where dengue is endemic. In such settings, patients are treated on clinical suspicion of dengue fever rather than after confirming the diagnosis, given the cost and inaccessibility of confirmatory tests. The CDS recruits all patients clinically suspected of dengue fever and then retrospectively confirms the diagnosis. Thus, we used data from all clinically suspected patients to develop and test the model ($DENV_{5f\text{-}AS}$) and then subsequently re-tested it in the subgroup of patients with confirmed diagnosis as a sensitivity analysis (model fairness) to demonstrate no significant difference in performance across these groups. The model also performed similarly when data were stratified by day of fever showing that it can be suitable for use for patients presenting symptoms at any time of the illness up to D3.

Previous studies that reported associations for adverse outcomes in dengue from Sri Lanka mostly used univariate analyses or logistic regression. These studies report associations with demographic factors such as age [16], clinical features such as abdominal pain [30], laboratory parameters such as liver transaminases, haematocrit, lymphocyte [20,31] and platelet count [16], and imaging findings such as gall bladder wall thickening on ultrasonography [32] to be associated with adverse outcomes. These studies are highly heterogenous and disregard the impact of individual patient observations given the reliance on traditional statistical methods which summarises dynamic variables as means, medians, or a single arbitrarily selected reading to represent a data array. We addressed this limitation by incorporating whole data arrays per variable, thus including dynamic variables observed across multiple instances per patient, to predict the outcome of interest. While the predictors of plasma leakage reported here are mostly similar that reported in previous studies which did not use machine learning, our observations re-confirm the importance of these risk predictors even when individual within- and between-host observations (granularity), missing data and non-linear associations are considered. Thus our observations further strengthen the evidence base for some risk predictors of plasma leakage. A single previous study from Sri Lanka attempted to use fuzzy logic fundamentals to predict the evolution of severe disease but it was limited to a small dataset was biased towards patients with adverse outcomes, 11 patients with dengue fever vs. 25 patients with dengue haemorrhagic fever [33].

Globally, there are ML studies for epidemiological risk prediction of dengue at community level [34–37] but only few have explored adverse outcomes at patient level. A Taiwanese study of 798 confirmed dengue patients used laboratory test results on admission and found that

NS1 antigenemia, age, anti-DENV IgG and IgM levels to be associated with severe dengue. However, this was a retrospective study which only reviewed case records which is unreliable in the accuracy of outcome classification compared to a prospective study. Furthermore, most variables identified as predictors in this study are not routinely available for clinicians in resource limited settings to effectively implement this model. A second retrospective Taiwanese study which analysed data from 4069 dengue patients found that liver transaminases and platelet counts within the first three days of fever to be associated with death [38]. There were no deaths in our cohorts to replicate these findings. Death is a rare outcome in dengue (<1%), even amongst hospitalised patients who have more severe disease [6]. Other studies used ML to predict adverse outcomes in dengue using genomics and transcriptomics data [39,40], the findings however are of limited value to clinicians managing patients for bed-side risk prediction.

On technical side, DENV$_{5f-AS}$ has key strengths. As such, it can provide predictions for incomplete instances even with up to 50% missingness. In addition, it is computationally inexpensive (no requirement of graphics processing units), open source, and can be run iteratively incorporating new patient data online or offline. It integrates RF and LightGBM taking advantage of both learning algorithms via average stacking, thus increasing model robustness. We used ensemble methods based on their enhanced robustness in predictions and feature selection compared to individual (base) models [41,42]. The post-prediction analysis of the abstention windows gave support to the robustness of the results. In addition, the clustering of SHAP values for model interpretability allowed a data-driven approach to identify distinct patterns of feature contributions and thereby providing an estimate on prediction certainty for new patients. This significantly improves the interpretability of feature contributions to the predictions providing insights to expected performance for unobserved patients. Improved AUC was observed as day of post fever onset increased (Table 3). The observation may be explained by the increasing number of available patients by day (Fig 1) as well as an indication that the predictors approach to more certain areas in the feature space for the prediction. As a simple alternative to ensemble models, decision tree from *rpart* package, performed poorly with AUC = 0.67±0.03 on the development set (and AUC = 0.64 on the test set) compared to the presented models. Current implementations of simpler models such as naïve bayes, OneR, and logistic regression could not handle missingness and thus were not used in this study. As an alternative to handle missingness, we used conditional multiple imputation from *mice* package in R and that did not improve the predictions (see S7 Table as compared to Table 3) where for example the AUC has dropped from 0.80 to 0.69. The imputation supported the notion that the feature selection was not markedly influenced by the missingness in the data.

This analysis used plasma leakage as the outcome of interest. Adverse outcomes in dengue are usually defined based on WHO clinical classifications published in 1997 and 2009. In the earlier classification more severe disease was categorised as dengue haemorrhagic fever (DHF). To be included in the DHF case definition four criteria had to be fulfilled (fever, evidence of bleeding, evidence of plasma leakage, platelet count <100,000/μl). Due to this stringent criterion some countries deviated from this classification in their local guidelines. For example, in Brazil, an intermediate severity category was defined for those not fulfilling all criteria [43] and in Sri Lanka DHF is mostly synonymous with plasma leakage in clinical practice. The later classification of WHO published in 2009 replaced the DHF classification with "severe dengue" which was characterised by either "severe" plasma leakage, "severe" organ dysfunction and "severe" haemorrhage. Again, given the subjective nature of this definition a model cannot reliably use this as an outcome if it needs to be used in a place other than where it was developed. Also, when criteria for severe dengue in the 2009 WHO classification is met the patients is likely to be in a critical condition with limited options for intervention. Ideally at-

risk patients should be identified prior to this stage. Considering all these factors we selected plasma leakage as the optimal outcome for risk prediction since it is a single criterion that can be objectively measured, and complications of dengue mostly (except for rare instances of abnormal bleeding) occur in patients who have plasma leakage. Given the dependence on WHO based classifications most dengue studies do not report incidence plasma leakage as a stand-alone figure but estimates from available literature suggests that 36.8% (95% CI: 35.4–38.2%) of all dengue patients have this outcome [6]. Thus, predicting this subgroup accurately can be a cost saver by reducing duration of hospital admissions. For example, in Sri Lanka which reported 105,409 cases of dengue in 2019 [44], an early discharge would have saved 18.02 USD per day per patient [11].

There are also limitations in this study to be focused for future improvements. Though 112 variables were available in the dataset some had to be excluded due to missing data. This is because in resource limited settings laboratory tests are performed only as needed. This is a reflection the real-world clinical data availability where this model will be implemented and hence cannot be considered a limitation of the dataset. Given the limitations, the model is still useful to select a subgroup of patients who are highly likely to develop plasma leakage for close monitoring. Data-driven stratification of patients could improve this as seen in our method based on the SHAP value clustering. It allowed identification of subgroups of patients on the test set and their distribution of feature values with higher or lower prediction performance compared with the overall performance evaluated on the test set. $DENV_{5f\text{-}AS}$ was developed using clinical data from Sri Lanka, making it relevant for this country with its homogenous population. The model could further be examined by external validations using similar datasets from other countries.

In summary, we utilised real world clinical data of five variables observed within the first 96 hours of fever in Sri Lankan patients with clinically suspected dengue fever to predict the likelihood of plasma leakage with PPV and NPV of 76.9% and 72.5%, respectively. Decision curve analysis provided evidence that the proposed model had superior net benefit for clinical utility than the available alternatives. This ML approach illustrated important implications that are relevant for model interpretability and robustness beyond merely predicting a disease outcome.

## Supporting information

**S1 Fig. Decision curve analysis, where x-axis indicates the probability threshold to classify to PL (predicted probability$\geq$ threshold) or noPL (otherwise), and y-axis indicates the net benefit limited to positive values.** The curves (decision curves) indicate the net benefit of the final model ($DENV_{5F\text{-}AS}$) as well as three alternatives (classifying no patient as PL, classifying all patients as PL, and random guess) over the threshold of 0 to 1, and net benefit defined as (true positives)/N–(false positive/N)× the odds at the probability threshold (N: number of samples). Vertical blue dash line indicates the selected cut-off value of 0.5 for the predictions. (DOCX)

**S2 Fig. Model interpretability—a) SHAP decision plots for $DENV_{5F\text{-}AS}$ on the test set in four clusters of SHAP values determined using the K-means method (k = 4).** The features are sorted from top to bottom by their mean absolute SHAP values (higher interpreted as more contributing). Each point represents an instance and the connected lines across features belong to the same instance. For each feature the points are scattered perpendicular to the horizontal line to minimise overlapping. Feature values are normalised to [0 1] by the min-max normalisation method and colour-coded (grey points are missing values), outliers were squished to the range using Hampel filter. The colour of each line is the same as the value of

the feature connected to in downwards direction. X-axis is the SHAP value computed for each instance. The order of the features is preserved from the original SHAP plot in Fig 4B) confusion matrix for each of the clusters for each matching cluster numbers (blue: all instances of the cluster, orange: earliest instance of patients in the cluster), the proportion of dengue and non-dengue patients are also captioned.
(DOCX)

**S1 Table. Full feature list and description.**
(DOCX)

**S2 Table. Filtered features (N = 43) for feature selection by Minimum Description Length (MDL) algorithm.**
(DOCX)

**S3 Table. Performance metrics on the test set using different number of selected features using Minimum Description Length (MDL) algorithm.**
(DOCX)

**S4 Table. Performance metrics on the test set using 10 different seeds based on "L'Ecuyer-CMRG" seeding in R version 4.1.2.**
(DOCX)

**S5 Table. Selected features using 10 different seeds.**
(DOCX)

**S6 Table. Summary statistics of feature values in each cluster.**
(DOCX)

**S7 Table. Performance metrics on the test set for a final model developed by the development set with imputed missingness using conditional multiple imputation from *mice* package seeds based on "L'Ecuyer-CMRG" seeding in R version 4.1.2.** The imputation resulted in the selection of HCT, HGB. Gender, age, and AST as selected features.
(DOCX)

**S8 Table. R package list.**
(DOCX)

**S1 Text. Supporting Information for the machine learning methods.**
(DOCX)

## Author Contributions

**Conceptualization:** Ramtin Zargari Marandi, Preston Leung, Chathurani Sigera, Daniel Dawson Murray, Deepika Fernando, Chaturaka Rodrigo, Cameron Ross MacPherson.

**Data curation:** Chathurani Sigera, Praveen Weeratunga, Deepika Fernando, Chaturaka Rodrigo, Senaka Rajapakse.

**Formal analysis:** Ramtin Zargari Marandi, Preston Leung, Chathurani Sigera.

**Investigation:** Ramtin Zargari Marandi, Preston Leung, Chaturaka Rodrigo.

**Methodology:** Ramtin Zargari Marandi, Preston Leung.

**Resources:** Praveen Weeratunga, Deepika Fernando, Chaturaka Rodrigo, Senaka Rajapakse.

**Software:** Ramtin Zargari Marandi.

**Supervision:** Chaturaka Rodrigo, Cameron Ross MacPherson.

**Validation:** Ramtin Zargari Marandi, Preston Leung, Chaturaka Rodrigo.

**Writing – original draft:** Ramtin Zargari Marandi, Preston Leung.

**Writing – review & editing:** Ramtin Zargari Marandi, Chathurani Sigera, Daniel Dawson Murray, Praveen Weeratunga, Deepika Fernando, Chaturaka Rodrigo, Senaka Rajapakse, Cameron Ross MacPherson.

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
