## [Decision Letter · Decision Letter 0]

2 Nov 2022

Dear Dr Zargari Marandi,

Thank you very much for submitting your manuscript "Development of a machine learning model for early prediction of plasma leakage in suspected dengue patients" for consideration at PLOS Neglected Tropical Diseases. As with all papers reviewed by the journal, your manuscript was reviewed by members of the editorial board and by several independent reviewers. In light of the reviews (below this email), we would like to invite the resubmission of a significantly-revised version that takes into account the reviewers' comments. 

Please make sure to respond to each one of the questions or comments made by the reviewers. We cannot make any decision about publication until we have seen the revised manuscript and your response to the reviewers' comments. Your revised manuscript is also likely to be sent to reviewers for further evaluation.

Sincerely,

Mabel Carabali, M.D., M.Sc., Ph.D.,

Academic Editor

Abdallah Samy

Section Editor

Please make sure to respond to each one of the questions or comments made by the reviewers.

Reviewer's Responses to Questions

**Key Review Criteria Required for Acceptance?**

**Methods**

-Are the objectives of the study clearly articulated with a clear testable hypothesis stated?

-Is the study design appropriate to address the stated objectives?

-Is the population clearly described and appropriate for the hypothesis being tested?

-Is the sample size sufficient to ensure adequate power to address the hypothesis being tested?

-Were correct statistical analysis used to support conclusions?

-Are there concerns about ethical or regulatory requirements being met?

Reviewer #1: Yes

Reviewer #2: There are some points that needs to be clarified, 

Regarding the training of the algorithms and the ensemble, how the algorithms parameters were tuned? How were the missing and unknown data handled?

Some variables were excluded due to their high cost such as viral load and data from ultrasonography, I wonder if the authors tried to train the algorithms including these variables to see how their presence in the dataset would affect the created model performance. 

Since there is no dataset for external validation, an internal-external validation approach could have been used [DOI: 10.1016/j.jclinepi.2015.04.005].

Reviewer #3: YES.

**Results**

-Does the analysis presented match the analysis plan?

-Are the results clearly and completely presented?

-Are the figures (Tables, Images) of sufficient quality for clarity?

Reviewer #1: Yes

Reviewer #2: Why the authors decided to go with an ensemble of both algorithms? Was the performance of the ensemble significantly better than the algorithms? I recommend comparing the performance of RF, LightGBM, and the ensemble to see if a model is statistically significantly better than the other [DOI: 10.1148/radiology.143.1.7063747]

Reviewer #3: YES.

**Conclusions**

-Are the conclusions supported by the data presented?

-Are the limitations of analysis clearly described?

-Do the authors discuss how these data can be helpful to advance our understanding of the topic under study?

-Is public health relevance addressed?

Reviewer #1: Yes

Reviewer #2: The model performance on the test set shows that the AUC improved when selecting patients by Day of post fever onset. I think this is an important clinical point that needs to be discussed more in the discussion.

Since the aim of the study is creating a model that can help clinicians in developing countries, I think that the model should have been implemented into some kind of a practical tool such as webapp or a nomogram or a stratification flow chart so it can be used in practice. 

In the discussion section, the authors mentioned that simpler models such as naïve bayes, OneR, and logistic regression could not handle missingness and thus were not used in this study. Does this mean that their performance were comparable to the used algorithms [RF and LightGBM]?

Reviewer #3: YES.

**Editorial and Data Presentation Modifications?**

Reviewer #1: (No Response)

Reviewer #2: (No Response)

Reviewer #3: (No Response)

**Summary and General Comments**

Reviewer #1: Summary 

The paper develops a machine learning model to predict plasma

leakage in suspected dengue patients using data from a prospective cohort

study in Sri Lanka. It is clinically important to identify predictors that

detect plasma leakage in the first few days dengue infection to improve

triage. A rigorous decision curve analysis is performed on the overall sam-

ple and by dengue diagnosis subgroup, and the model is interpreted using

Shapley additive explanations.

The authors may consider new analyses for (1) a multi-label classifica-

tion task (i.e., PL, noPL, not recorded); and (2) retaining variables with

missing values (that are not completely missing in the training set), or im-

puting these values prior to classification. These suggestions are explained

below.

Major comments

1. Why was plasma leakage not recorded in 172 patients? As reported

in Table 1, there is a much higher proportion of missing outcome

values in the non-dengue patients (26%) compared to the dengue

patients (16%), which suggests the missing values are not missing at

random and may indicate information of clinical importance. In this

case, it might be more realistic to treat the prediction problem as a

multi-class problem (i.e., PL, noPL, not recorded).

2. I’m unsure of the claim that removing instances with more than 50%

of the features missing would reduce biased interpretation of feature

contributions. Removing these instances would almost surely bias the

interpretation because the missing values are presumably not missing

at random. There are generally higher proportions of missing feature

values among the non-dengue patients compared to the dengue pa-

tients (Table 1), presumably because the tests are given more to the

sicker patients. I expect that excluding these instances, rather than

let the algorithms handle the missing data internally or impute the

values, would limit the ability of the classifiers to learn the task.

Several papers show that imputing the missing values (e.g., with k-

NN) can outperform the internal methods used by decision tree-based

algorithms to treat missing data (e.g., https://www.tandfonline.

com/doi/pdf/10.1080/08839514.2018.1448143). This paper shows

that adding missing-data perturbation prior to imputation can actu-

ally improve prediction accuracy in supervised classification tasks by

regularizing the classifier.

3. Relatedly, in the discussion two contradictory statements are made:

that the proposed classifier can handle missing data and later, that

many variables had be excluded due to missingness. I can see why

variables that are completely missing in the training set (Dengue

virus serotype and viral load) need to be dropped, but not understand

the intuition for dropping any other variables.

Minor comments

1. Pg. 4, typo: “focusses”

2. Pg. 5: I don’t understand why the viral load and ultrasonography

predictors are removed due to high costs. Is this because the non-

dengue patients don’t get tested? Why are the ultrasound predictors

not summarized in Table 1?

3. Table 2: how is the variance of the performance metrics calculated?

4. Figure 3a.: Should the x-axis be labeled instead “1-Specificity” (the

FPR)?

Reviewer #2: The study aims to create a machine learning based model that can predict the occurrence of plasma leakage in patients with dengue fever based on accessible clinical and lab parameters. The study is well designed and of a huge clinical importance especially when thinking about implying machine learning models in order to decrease costs in developing countries. 

There are some points that needs to be clarified, 

Regarding the training of the algorithms and the ensemble, how the algorithms parameters were tuned? How were the missing and unknown data handled?

Some variables were excluded due to their high cost such as viral load and data from ultrasonography, I wonder if the authors tried to train the algorithms including these variables to see how their presence in the dataset would affect the created model performance. 

Since there is no dataset for external validation, an internal-external validation approach could have been used [DOI: 10.1016/j.jclinepi.2015.04.005]. 

The reference [18] is focusing on the random survival forests which is mainly used in survival analysis. I would recommend replacing it with a more-focusing reference on random forests in binary classification.

Why there is a difference between the number of instances and number of patients? Does this mean that some patients had multiple instances?

Why the authors decided to go with an ensemble of both algorithms? Was the performance of the ensemble significantly better than the algorithms? I recommend comparing the performance of RF, LightGBM, and the ensemble to see if a model is statistically significantly better than the other [DOI: 10.1148/radiology.143.1.7063747]

The model performance on the test set shows that the AUC improved when selecting patients by Day of post fever onset. I think this is an important clinical point that needs to be discussed more in the discussion.

Since the aim of the study is creating a model that can help clinicians in developing countries, I think that the model should have been implemented into some kind of a practical tool such as webapp or a nomogram or a stratification flow chart so it can be used in practice. 

In the discussion section, the authors mentioned that simpler models such as naïve bayes, OneR, and logistic regression could not handle missingness and thus were not used in this study. Does this mean that their performance were comparable to the used algorithms [RF and LightGBM]?

Reviewer #3: In the manuscript entitled "Development of a machine learning model for early prediction of plasma leakage in

suspected dengue patients" by Marandi et al., the authors describe their work to develop a tool for plasma leakage prediction in the setting of dengue fever. This is a well performed study, with an adequate methodology. Unfortunatelly, the tool had not a high sensitivity in the prediction of plasma leakage; however, it showed to have a high specificity for that purpose. Therefore, although this is not a good tool to be taken into consideration alone in the decision of discharging patiens from hospitals, it may be useful to aid health care professionals to detect those patients who demand special vigilance due to the risk of clinical deterioration. Moreover, that tool does not require expensive diagnostic tests. These issues make the machine learning model developed by the authors a very useful tool specially in low-income countries affected by dengue virus, since the prediction of serious complications in patients infected with the virus is a challenging work for health providers.

PLOS authors have the option to publish the peer review history of their article (what does this mean?). If published, this will include your full peer review and any attached files.

Reviewer #1: No

Reviewer #2: No

Reviewer #3: No
---

## [Decision Letter · Decision Letter 1]

1 Feb 2023

Dear Dr Zargari Marandi,

Thank you very much for submitting your manuscript "Development of a machine learning model for early prediction of plasma leakage in suspected dengue patients" for consideration at PLOS Neglected Tropical Diseases. As with all papers reviewed by the journal, your manuscript was reviewed by members of the editorial board and by several independent reviewers. The reviewers appreciated the attention to an important topic. Based on the reviews, we are likely to accept this manuscript for publication, providing that you modify the manuscript according to the review recommendations. 

Thank you so much for providing a revised version of the manuscript. However, some aspects still need to be clarified before considering it for publication.

The document highlights an interesting method and novel approach. Despite the favorable review, the manuscript still needs to tone down some of the conclusions and statements while acknowledging some of the limitations.

For instance, the missing data is an issue, yes it is a "real world" data, but analytically there are several approaches that could be used to address the missing data problem. Authors should indicate how, this would affect the results and their interpretations. Especially, plasma leakage was missing overall 20% but 25% for not dengue and 15% for dengue cases. 

- Could you please comment on the effect of the differential missing for the outcome? 

- Authors indicated that "DENV5f-AS can handle missing data", could you please provide further details on the MICE implementation and how the S7 table compare to the main results in the main text?

Also, overall results and implications from this ML exercise are not different from standard/traditional regression approaches with predictive approaches. This is, looking at hematological parameters and liver enzimes as predictors of severe dengue and plasma leakage has been the standard for the last several decades. My point, highlighting the effort of the authors, is to acknowledge that although you are contributing to the knowledge by illustrating the use of the novel methods, results are confirming some of the already stablished knowledge on the field. This should be addressed in the abstract, author summary, discussion and conclusion. Especially, because the results do not support that you provide "an artificial intelligence solution to predict plasma leakage in resource limited settings". The results showed that using an artificial intelligence tool, such ML, and with data from a resource limited setting, the authors observed a consistent prediction of plasma leakage by several known factors. These results are still the result from prediction, and in clinical settings, clinicians will only use the results form the literature but not necessarily recalculate these parameters using the data available to decide the course of treatment for a given patient under a given clinical presentation. To improve the quality of the literature we are getting out there for our audience, we must acknowledge and address our limitations accordingly.

Additional points: 

- Please provide the % next to the missing data figures in table 1.

- Please indicate how would you interpret the p-values for the False discovery rate correction for multiple testing in this context?

- Also, consider removing the p-values altogether or including them in the supplementary tables.

- Authors summary starts with "We queried PubMed and Google Scholar for any publications between 1990 and 2022 using search terms such as “machine learning” or “artificial intelligence” or “prediction” and “plasma leakage”. which give the impression we are about to read s Systematic review, please consider rephrasing the section.

Sincerely,

Mabel Carabali, M.D., M.Sc., Ph.D.,

Academic Editor

Abdallah Samy

Section Editor

Thank you so much for providing a revised version of the manuscript. However, some aspects still need to be clarified before considering it for publication.

The document highlights an interesting method and novel approach. Despite the favorable review, the manuscript still needs to tone down some of the conclusions and statements while acknowledging some of the limitations.

For instance, the missing data is an issue, yes it is a "real world" data, but analytically there are several approaches that could be used to address the missing data problem. Authors should indicate how, this would affect the results and their interpretations. Especially, plasma leakage was missing overall 20% but 25% for not dengue and 15% for dengue cases. 

- Could you please comment on the effect of the differential missing for the outcome? 

- Authors indicated that "DENV5f-AS can handle missing data", could you please provide further details on the MICE implementation and how the S7 table compare to the main results in the main text?

Also, overall results and implications from this ML exercise are not different from standard/traditional regression approaches with predictive approaches. This is, looking at hematological parameters and liver enzimes as predictors of severe dengue and plasma leakage has been the standard for the last several decades. My point, highlighting the effort of the authors, is to acknowledge that although you are contributing to the knowledge by illustrating the use of the novel methods, results are confirming some of the already stablished knowledge on the field. This should be addressed in the abstract, author summary, discussion and conclusion. Especially, because the results do not support that you provide "an artificial intelligence solution to predict plasma leakage in resource limited settings". The results showed that using an artificial intelligence tool, such ML, and with data from a resource limited setting, the authors observed a consistent prediction of plasma leakage by several known factors. These results are still the result from prediction, and in clinical settings, clinicians will only use the results form the literature but not necessarily recalculate these parameters using the data available to decide the course of treatment for a given patient under a given clinical presentation. To improve the quality of the literature we are getting out there for our audience, we must acknowledge and address our limitations accordingly.

Additional points: 

- Please provide the % next to the missing data figures in table 1.

- Please indicate how would you interpret the p-values for the False discovery rate correction for multiple testing in this context?

- Also, consider removing the p-values altogether or including them in the supplementary tables.

- Authors summary starts with "We queried PubMed and Google Scholar for any publications between 1990 and 2022 using search terms such as “machine learning” or “artificial intelligence” or “prediction” and “plasma leakage”. which give the impression we are about to read s Systematic review, please consider rephrasing the section.

Reviewer's Responses to Questions

**Key Review Criteria Required for Acceptance?**

**Methods**

-Are the objectives of the study clearly articulated with a clear testable hypothesis stated?

-Is the study design appropriate to address the stated objectives?

-Is the population clearly described and appropriate for the hypothesis being tested?

-Is the sample size sufficient to ensure adequate power to address the hypothesis being tested?

-Were correct statistical analysis used to support conclusions?

-Are there concerns about ethical or regulatory requirements being met?

Reviewer #1: (No Response)

Reviewer #2: Yes

**Results**

-Does the analysis presented match the analysis plan?

-Are the results clearly and completely presented?

-Are the figures (Tables, Images) of sufficient quality for clarity?

Reviewer #1: (No Response)

Reviewer #2: Yes

**Conclusions**

-Are the conclusions supported by the data presented?

-Are the limitations of analysis clearly described?

-Do the authors discuss how these data can be helpful to advance our understanding of the topic under study?

-Is public health relevance addressed?

Reviewer #1: (No Response)

Reviewer #2: Yes

**Editorial and Data Presentation Modifications?**

Reviewer #1: (No Response)

Reviewer #2: (No Response)

**Summary and General Comments**

Reviewer #1: I am satisfied with the authors' responses. In particular, the addition of analyses on a multiply imputed dataset (S7 Table) and the inclusion of a link to the API for the final model greatly improves the paper.

Reviewer #2: The authors have addressed all comments

PLOS authors have the option to publish the peer review history of their article (what does this mean?). If published, this will include your full peer review and any attached files.

Reviewer #1: No

Reviewer #2: No

Figure Files:

Data Requirements:

Reproducibility:

References

---

## [Decision Letter · Decision Letter 2]

24 Feb 2023

Dear Dr Zargari Marandi,

We are pleased to inform you that your manuscript 'Development of a machine learning model for early prediction of plasma leakage in suspected dengue patients' has been provisionally accepted for publication in PLOS Neglected Tropical Diseases.

Best regards,

Mabel Carabali, M.D., M.Sc., Ph.D.,

Academic Editor

Abdallah Samy

Section Editor

Reviewer's Responses to Questions

**Key Review Criteria Required for Acceptance?**

**Methods**

-Are the objectives of the study clearly articulated with a clear testable hypothesis stated?

-Is the study design appropriate to address the stated objectives?

-Is the population clearly described and appropriate for the hypothesis being tested?

-Is the sample size sufficient to ensure adequate power to address the hypothesis being tested?

-Were correct statistical analysis used to support conclusions?

-Are there concerns about ethical or regulatory requirements being met?

Reviewer #1: (No Response)

Reviewer #2: (No Response)

**Results**

-Does the analysis presented match the analysis plan?

-Are the results clearly and completely presented?

-Are the figures (Tables, Images) of sufficient quality for clarity?

Reviewer #1: (No Response)

Reviewer #2: (No Response)

**Conclusions**

-Are the conclusions supported by the data presented?

-Are the limitations of analysis clearly described?

-Do the authors discuss how these data can be helpful to advance our understanding of the topic under study?

-Is public health relevance addressed?

Reviewer #1: (No Response)

Reviewer #2: (No Response)

**Editorial and Data Presentation Modifications?**

Reviewer #1: (No Response)

Reviewer #2: (No Response)

**Summary and General Comments**

Reviewer #1: I've reviewed the authors' responses to the Editor's comments and changes made to the paper and am satisfied with the paper in its current form.

Reviewer #2: (No Response)

PLOS authors have the option to publish the peer review history of their article (what does this mean?). If published, this will include your full peer review and any attached files.

Reviewer #1: No

Reviewer #2: No

---

## [Editor Report · Acceptance letter]

9 Mar 2023

Dear Dr Zargari Marandi,

We are delighted to inform you that your manuscript, "Development of a machine learning model for early prediction of plasma leakage in suspected dengue patients," has been formally accepted for publication in PLOS Neglected Tropical Diseases.

Best regards,

Shaden Kamhawi

co-Editor-in-Chief

Paul Brindley

co-Editor-in-Chief
